# Effect of Heat Treatment on Microstructure and Thermal Conductivity of Thermal Barrier Coating

**DOI:** 10.3390/ma14247801

**Published:** 2021-12-16

**Authors:** Kyomin Kim, Woochul Kim

**Affiliations:** School of Mechanical Engineering, Yonsei University, Seoul 03722, Korea; kyomkim@yonsei.ac.kr

**Keywords:** thermal barrier coating, heat treatment, thermal conductivity

## Abstract

Thermal barrier coatings (TBCs) are essential for increasing the inlet temperature of gas turbines to improve their thermal efficiency. Continuous exposure to flames is known to affect the thermal properties of TBCs, degrading the performance of gas turbines as a consequence. In this study, we quantified the changes in the thermal conductivity of yttria-stabilized zirconia coatings with respect to various heat treatment temperatures and times. The coating exhibited an increase in thermal conductivity after heat treatment, with higher heat treatment temperatures resulting in greater thermal conductivity. The coatings were analyzed by X-ray diffraction and scanning electron microscopy before and after heat treatment. Results showed that there was little change in thermal conductivity due to phase changes and grain size. We conclude that pore structures, i.e., circular and lamellar pores, affected the change in thermal conductivity. Specifically, we confirmed that the change in thermal conductivity depends on the size of the lamellar pores.

## 1. Introduction

The development of thermal barrier coatings (TBCs) has improved the durability of gas turbine components exposed to high temperatures and has improved the energy efficiency of gas turbines by increasing the turbine inlet temperature [1]. However, the temperature of the turbine inlet cannot be increased indefinitely due to the maximum operating temperature of the superalloy. When the surface temperature exceeds 900 °C, the nickel-based superalloy used as the substrate is oxidized and corrodes [2]. Therefore, low thermal conductivity is essential for TBCs, to prevent oxidation and corrosion of superalloys at high turbine inlet temperatures. To lower the thermal conductivity of superalloys, several studies on structural changes [3,4] and manufacturing using rare earths [5] have been conducted. In general, TBCs are composed of a ceramic top coating that is directly exposed to a high-temperature environment and a metal alloy bond coating for oxidation resistance. Yttria-stabilized zirconia (YSZ) is a commonly used top coating because it is capable of maintaining a constant low thermal conductivity even at high temperatures [5]. An increase in the yttria content results in lower thermal conductivity of the TBC. However, the amount of yttria content should be within an appropriate range because it can affect the thermal cycle stability. The highest stability is shown when the yttria content is 7 wt% [6]. Therefore, 7YSZ is widely used as a material for TBC.

Among the various coating technologies, air plasma spray (APS) is the most common method used for TBCs and involves injecting ceramic powder into a high-temperature plasma to make it molten before adhering it to a metal substrate [7]. Due to the presence of unmelted particles and structural defects such as interlayer pores and microcracks caused by the layered structure, APS based coatings have a low thermal conductivity in the range of 0.8–1.1 W/m-K [8]. It is known that the durability of coatings decreases with increasing operation time. To improve the structural characteristics of the coatings, low-pressure plasma spraying can be used to improve bonding strength and to decrease porosity, while vacuum plasma spraying can minimize the formation of oxides [9]. Electron-beam physical vapor deposition (EB-PVD) is another common coating method. An electron beam is irradiated on the surface of a sintered material made of ceramic powder in a high vacuum state to form gaseous particles. The particles are then deposited in a columnar manner on the surface of a substrate material [10]. EB-PVD coating results in a relatively high thermal conductivity of 1.5–1.9 W/m-K, due to the presence of different structures from those formed by plasma spraying (PS) [8]. EB-PVD, which is mainly used in a similar way to APS, shows high stability, but has a complicated manufacturing process and exhibits relatively high thermal conductivity compared to APS [11]. On the other hand, the APS manufacturing process is simple.

Because TBCs are continuously exposed to high temperatures, it is important to understand the changes in their material properties in order to predict their lifetime. Many researchers have studied the effect of sintering caused by high temperature exposure on changes in microstructures. Zambrano et al. [12] and Osorio et al. [13] analyzed the porosity of YSZ manufactured with APS after heat treatment at 1100 °C for 1700 h. After annealing, the porosity decreased due to the sintering effect. Similarly, Ekberg et al. [14] controlled the spray parameters and heat-treated the 4YSZ deposited by suspension plasma spraying at 1150 °C for 50 and 200 h. The decrease in porosity due to the sintering effect was confirmed through porosity analysis. Thompson et al. [15] observed the microstructure of heat-treated 8YSZ with APS at 1000–1400 °C. The microcracks inside the coating were healed by heat treatment. Ilvasky et al. [16] examined heat-treated YSZ fabricated by PS at 600–1400 °C. They found that the surface area of pores and cracks inside the coating depended on the heat treatment temperature. Furthermore, many studies show a decrease in porosity due to sintering [17,18,19,20]. These studies confirm that the microstructure of TBCs changes due to the sintering effect of heat treatment. Although the relationship between heat treatment and thermal conductivity has been studied, the exact cause of the change in thermal conductivity due to the heat treatment temperature and time is not known. Moreover, it is difficult to know whether the heat treatment temperature or time is important in changing the microstructure of TBCs.

Thermal conductivity is an important property that determines the performance of TBCs. Various studies on factors affecting the thermal conductivity of TBCs have been reported. Kabacoff et al. [21] found that reducing the grain size of TBCs reduced their thermal conductivity. Raghavan et al. [22] reported that there was no change in thermal conductivity due to grain boundaries in YSZ prepared by APS using nanocrystalline powder with grain sizes ranging from 70 to 400 nm. Markocsan et al. [23] observed the microstructure of various materials containing 7-8YSZ that were manufactured by controlling the power and feed rate of APS, and experimentally demonstrated that flat pores cause a greater reduction in thermal conductivity than do spherical pores. In addition, Huang et al. [24] reported that the shape of the pores inside TBCs manufactured by APS affect the resultant change in thermal conductivity. Chen et al. [25] manufactured 4YSZ coatings with different sizes and structures using APS; they found that the thermal conductivity of the coating was affected by the porosity and grain boundary density at low temperatures and by changes in monoclinic zirconia (m-ZrO_2_) at high temperatures. Previous studies have established that the thermal conductivity of TBCs is affected by grain size [21,22], pore structure [23,24] and phase changes [25]. It is necessary to understand the factors that change the thermal conductivity based on the working temperature and time of the TBCs, such as the change in microstructure due to the sintering effect according to the heat treatment conditions. Therefore, it is important to know the changes in grain size, pore structure and phase change due to heat treatment, as they affect the thermal conductivity of TBCs.

Numerous studies have investigated the relationship between heat treatment and thermal conductivity. It is known that the thermal conductivity of TBCs increases after heat treatment [26,27]. Additionally, the change in thermal conductivity depends on the exposure temperature and time [28,29]. Furthermore, there are various causes for changes in thermal conductivity due to heat treatment. Limarga et al. [30] demonstrated an increase in thermal conductivity due to grain growth after heat treatment at 1200 and 1400 °C using fully dense YSZ fabricated by spark plasma sintering; the effect of porosity was excluded. Kai et al. [31] examined a significant increase in the thermal conductivity of nanostructured YSZ coating due to changes in grain size. Trice et al. [32] confirmed that the thermal conductivity of 7YSZ coating manufactured by PS was higher than that of the as-sprayed coating after heat treatments at 1000, 1200, and 1400 °C for 50 h. This was caused by the closure of microcracks between the layers, the sintering of porosity and an increase in m-ZrO_2_ content with high thermal conductivity according to the heat treatment temperature. Moreover, researchers reported that the porosity of heat-treated coatings can affect thermal conductivity [33,34]. These studies examined the effects of grain growth, phase change and porosity caused by heat treatment on the thermal conductivity of TBCs. Several studies have shown that annealing temperature or time causes a change in the thermal conductivity of TBCs. However, there are insufficient studies on the effect of heat treatment temperature and time on changes in the thermal conductivity of TBCs. There is an insufficient amount of research to determine whether the dominant effect on the change in thermal conductivity of TBCs due to heat treatment is temperature or time.

This study aims to investigate the effects of thermal aging and pore structure on the thermal conductivity of TBCs. First, 7YSZ coatings manufactured by APS were investigated. The coatings were then heat-treated at various temperatures for different lengths of time. After heat treatment, the samples were analyzed using X-ray diffraction (XRD) and scanning electron microscopy (SEM). Based on the analyzed date, the microstructures of the samples were characterized using an image-analysis program and the relationship between the thermal conductivity and the microstructure of the heat-treated samples was elucidated.

## 2. Experimental Procedure

The TBC samples were prepared by APS, and consisted of a top coating of 7YSZ (ZRO-270-4, Praxair Surface Technologies, Indianapolis, IN, USA), a bond coating of CoNiCrAlY (Amdry 9951, Sulzer Metco, Westbury, NY, USA) and a Ni-based superalloy CM247LC substrate. The material used for the top coating was a powder containing 7 wt% of Y_2_O_3_ based on ZrO_2_. The powder was agglomerated and sintered, resulting in an average size of about 11 μm. A schematic of the fabrication method is shown in Figure 1. The top coating deposited with APS was sprayed on the substrate at 600 A and 80 V. The speed at which the top coating was sprayed on the base material was 300–600 mm/s. The spraying distance was about 76–127 mm. The 7YSZ powder was supplied to the injector and deposited on the substrate through a spray stream. The surface of the substrate was sandblasted before spraying to increase the contact area and roughness. As shown in the dotted circle in Figure 1, the powder was randomly deposited on the adhesive surface in the form of splats. Pores were generated between the deposited powder particles. Pore size depends on the power of the spray gun, the size and amount of the powder and the deposition distance [35]. Air becomes trapped inside the pores, reducing thermal conductivity. To ensure a stable thermal cycle, the thickness of the top and bond coatings was set to approximately 400 and 250 μm, respectively. The bond coating plays a crucial role, as it enhances the adhesion of the top coating [36], relieves the thermal stress caused by the difference in thermal expansion coefficient between the substrate and the ceramic top coating material [37] and improves oxidation resistance and hot corrosion resistance at high temperature [38].

To observe the microstructure of the samples, the coated substrate was cut, mounted, ground and polished, and its cross-section was analyzed by field emission SEM (IT-500HR, JEOL, Tokyo, Japan). A cross-section of the TBC is shown in Figure 2. The top coating, bond coating and substrate layers are shown in Figure 2a. The thermally grown oxide (TGO) that protects the superalloy from oxidation is shown in Figure 2 (inset). The top coating was approximately 400 μm thick, and pores of various sizes and shapes were distributed in the sample. Figure 2b shows an enlarged area of Figure 2a. Because the powder was adhered to the bond coating by high-speed plasma and was accumulated in a layered structure, pores formed by sparse deposition between powder particles, and unmelted particles that did not receive sufficient heat can be identified. Two types of pores, namely lamellar and circular pores, are present. Most of the circular pores were small (<2 μm^2^). In addition, horizontal lamellar pores, mainly caused by the layered structure, were observed.

Thermal aging, which simulates a coating being exposed to high temperatures for a long time, was performed in a furnace at 1150 and 1200 °C to examine the effect of changes in thermal conductivity at high temperatures. These thermal aging temperatures were chosen because they are close to the working temperature (1050–1200 °C) at which most YSZ is used [24]. Five different aging times were set to evaluate the change in thermal conductivity with exposure time. The thermal aging times of 25, 75, 125, 225 and 425 h were considered at 1150 °C, whereas, at 1200 °C, 10, 25, 50, 100 and 175 h were used. These values are based on the exfoliation of the top and bond coatings by TGO growth due to thermal aging [39]. After thermal aging, natural cooling was performed to prevent exfoliation of the coatings due to rapid cooling caused by the difference in the coefficient of thermal expansion between the coating and the substrate. The presence of 7YSZ in the as-sprayed and thermally aged coatings was determined using high-resolution XRD (SmartLab, Rigaku, Tokyo, Japan), as well as by analyzing whether any phase change occurred after thermal aging. In addition, the surfaces and cross-sections of the samples were observed via SEM to examine their microstructural changes.

To measure the thermal conductivity of the YSZ coating, the samples were soaked in 37% HCl to dissolve the oxide layer between the bond coating and the top coating. The separated YSZ coating was thoroughly dried in the furnace after washing with deionized (DI) water (Daejung Chemicals & Metals Co., Siheung, Korea). The thermal conductivity of a coating can be expressed as *k* = *αρ**C_p_*, where *k* is the thermal conductivity, *α* is the thermal diffusivity, *ρ* is the density of YSZ and *C_p_* is the specific heat.

The thermal diffusivity of the specimens was measured using a laser flash apparatus (LFA 457 MicroFlash, NETZSCH, Selb, Germany). A laser pulse is irradiated on one side of the specimen and the heat transferred through the specimen is read by a detector on the other side. Thermal diffusivity is expressed as half of the maximum time of the temperature rise on the backside and thickness of the specimen. Because YSZ coating is transparent at high temperatures, a graphite coating of a few micrometers was sprayed to increase the absorption of the irradiated surface and the emission of the detection surface. To measure the change in thermal conductivity according to the measurement temperature, the measurement range was set from 100 to 1100 °C, with 100 °C increments. The density was measured by measuring the mass of the coating in air and water using the Archimedes’ method.

## 3. Results and Discussion

Figure 3 shows the XRD spectra of the samples after thermal aging at 1200 °C. The coating of the as-sprayed sample exhibited the same peaks determined by Zhang et al. [40], which confirms that the YSZ coating used in this experiment had the same composition as that used in the reference literature. Although the 1200 °C samples show the same peak intensity as the reference literature, intensities are weaker compared to that of the as-sprayed samples results as shown in Figure 3 (inset). This difference is caused by the change in texture through heat treatment [41], since heat treatment temperature affects intensity. YSZ consists of tetragonal (t-ZrO_2_) and cubic (c-ZrO_2_) zirconia phases. Some heat treatment conditions also generate m-ZrO_2_ [32]. Because m-ZrO_2_ has a higher thermal conductivity than t-ZrO_2_ or c-ZrO_2_, the thermal conductivity increases with the generation of m-ZrO_2_. However, because all samples in Figure 3 exhibit the same peaks and consist of t-ZrO_2_ and c-ZrO_2_, m-ZrO_2_ is not present and the change in the thermal conductivity is not due to phase change.

Electrons, lattice vibration and radiation contribute to the thermal conductivity of a solid [42]. Because zirconia-based alloy is an electrically insulating material, its thermal conductivity is not affected by electrons, but by lattice vibration. Thermal conductivity is affected by heat capacity, phonon group velocity and mean free path [5]. The thermal conductivity due to lattice vibration can be expressed as *k = 1/3 C_v_vl*, where *C_v_* is the heat capacity, *v* is the phonon group velocity and *l* is the mean free path of phonons. To change the thermal conductivity of a material, variations in *C_v_*, *v* and *l* are required. However, heat capacity is negligible above the Debye temperature. In the case of zirconia, the Debye temperature is 380 K; therefore, the changes in the specific heat above the Debye temperature are negligible [43]. Phonon group velocity can be expressed as *v* = (*E*/*ρ*)^1/2^ where *E* is the elastic modulus and *ρ* is the density of YSZ. The elastic modulus of YSZ increases as the heat treatment continues, and at high-heat treatment temperatures the rate of increase is high [15,44,45]. Kim et al. [44] reported that the change in the elastic modulus of YSZ was due to pores or cracks inside the coating. There is no significant difference in the density of YSZ with respect to heat treatment conditions [46]. The mean free path is not affected at grain boundaries larger than several hundred nanometers [21]. On the other hand, the elastic modulus that affects the phonon group velocity is changed by pores or cracks. Therefore, we suspect that thermal conductivity after heat treatment is caused by changes in pores and cracks, which could affect the elastic modulus.

Figure 4 shows the thermal conductivity of all TBC samples with respect to their measured temperature. Thermal conductivity is inversely proportional to temperature; this is known as the phonon–phonon scattering effect [5]. As the temperature increases, the thermal conductivity converges to a plateau. It can be confirmed that the heat-treated samples exhibit higher thermal conductivity than their as-sprayed counterparts, and that heat treatment at 1200 °C resulted in higher thermal conductivity than that at 1150 °C. According to previous studies, the change in thermal conductivity due to heat treatment is caused by a decrease in porosity due to the sintering effect [28,32]. Additionally, the decrease in porosity due to thermal annealing depends on the heat treatment temperature. The heat-treated sample at high temperature had a higher thermal conductivity than the sample heat-treated at low temperature because the porosity was significantly reduced. Additionally, the porosity of the sample heat-treated at the same heat treatment temperature decreased as the heat treatment time increased [12,13]. Since the decrease in porosity increases the thermal conductivity, the thermal conductivity increases as the heat treatment time increases. As shown in Figure 4, the sample heat-treated at 1200 °C has a higher thermal conductivity than the sample heat-treated at 1150 °C. In addition, the thermal conductivity increases as the heat treatment time increases at the same heat treatment temperature. The material properties and microstructure were analyzed to determine the cause of the significant change in thermal conductivity when the heat treatment temperature is 1200 °C compared to 1150 °C. To understand the factors affecting the change in thermal conductivity according to the thermal aging conditions, the density, grain size and porosity of the samples were analyzed.

The theoretical density of 7YSZ is 6050 kg/m^3^. However, APS-fabricated YSZ contains numerous pores, and the measured density was approximately 5600–5800 kg/m^3^. Figure 5a shows the relative density of the TBC specimens. In general, the densification of the material causes an increase in thermal conductivity after thermal aging. Likewise, the relative density increases as heat treatment continues—and at higher-heat treatment temperatures—because densification occurs due to the reduction of internal pores.

The grain sizes after thermal aging are shown in Figure 5b. The grain size of the YSZ was measured with an image analysis software (ImageJ, NIH, MA, USA) using a SEM image. The SEM image can be converted into pixels using ImageJ to measure the size of the pixels. After calculating the grain area on the coating surface with ImageJ, the diameter of the circle with its area equal to the grain area was referred to as the grain size. For example, an image of the sample that was thermally aged at 1200 °C for 25 h was captured (Figure 6a), the image was analyzed using ImageJ (Figure 6b) and a histogram of the grain size was plotted (Figure 6c). Phonons scattered by grain boundaries cause a decrease in thermal conductivity. Grain size is closely related to thermal conductivity. Overall, the grains had an average size of 200–300 nm and a standard deviation of 50–100 nm. The grain size and thermal conductivity of the coating after thermal aging increased compared to that of the as-sprayed coating. When the nanosized grains are on the same order as the mean free path of YSZ, the thermal conductivity is significantly reduced [21]. However, when the grain size of dense YSZ is between 70 and 400 nm, the change in thermal conductivity due to boundary scattering is insignificant [22]. In general, when the mean free path causes changes in thermal conductivity due to the influence of the grain boundaries, the mean free path is comparable to the grain size at low temperatures. Therefore, a grain size of approximately 250 nm does not affect the change in thermal conductivity after heat treatment.

Figure 7a,b show graphs of the porosity and pore size of the sample cross-sections, respectively. Lamellar and circular pores contributed to the total porosity. The total porosity of the heat-treated samples was lower than that of the as-sprayed sample. When the heat treatment temperature was increased from 1150 °C to 1200 °C, the lamellar porosity decreased but the circular porosity increased. A high annealing temperature affects the transformation of lamellar pores into circular pores [47]. Figure 7b shows the average sizes of the lamellar and circular pores. The difference in lamellar pore size according to the heat treatment conditions was consistent with the change in thermal conductivity. In contrast, the circular pores were all similar in size (<1 μm^2^). Circular pores with areas 2 μm^2^ or lower have no significant effect on thermal conductivity; therefore, they can be excluded from this analysis [24]. However, lamellar pores are known to significantly affect thermal conductivity [24,48,49]. The lamellar pores between the coating layers reduce the thermal conductivity because they have a lower heat flux than the perfectly contacted coating layers [50]. Vertical lamellar pores increase the thermal fatigue durability of the coating [39]. However, because they occupy an extremely small area and their contribution to heat flux change is small, the change in thermal conductivity due to vertical lamellar pores was excluded. Figure 7c–e shows cross-sections of the as-sprayed and heat-treated samples (1150 °C for 125 h and 1200 °C for 50 h). Arrows indicate lamellar pores present in the cross-section. After heat treatment, the lamellar pore size was significantly reduced. After heat treatment at 1200 °C the lamellar pores became more circular. Therefore, the increase in thermal conductivity after heat treatment was due to the decrease in the size of the lamellar pores. In addition, it was confirmed that the increase in thermal conductivity at a high-heat treatment temperature was caused by lamellar pores becoming circular pores.

## 4. Conclusions

The change in thermal conductivity of 7YSZ coating manufactured by APS was analyzed with respect to several heat treatment conditions. The coatings were heat-treated at 1150 and 1200 °C, and their thermal conductivity increased as the heat treatment temperature increased. Factors that can affect the thermal conductivity of the samples (phase change, grain size, density, pores) were observed. After heat treatment, phase changes were not detected, and grain size (±16%) and density (±0.9%) exhibited similar values to that of the as-sprayed samples; it was found that these factors have little effect on thermal conductivity. The coating contained lamellar (60%) and circular pores (40%). Lamellar pores have a dominant influence on the change in thermal conductivity, whereas circular pores have less influence. Through porosity analysis using an image analysis program, it was observed that the average lamellar pore size was reduced to 33% at 1150 °C for 125 h and 62% at 1200 °C for 50 h compared to the as-sprayed sample. Accordingly, the thermal conductivity values increased by 21% and 119%, respectively. Heat treatment contributed to the transformation of lamellar pores into circular pores. The higher heat treatment temperature resulted in a higher thermal conductivity because the size of the lamellar pores was further reduced. As a result, because TBCs are continuously exposed to high temperatures, their thermal conductivity increases, which in return degrades their turbine performance. The measured results have important implications as to why the thermal conductivity of TBCs change under continuous exposure to high temperatures. Long-life coatings designed based on these results are expected to be applicable to high turbine inlet temperatures and in return to further increase the efficiency of the turbine.

## Figures and Tables

**Figure 1 materials-14-07801-f001:**
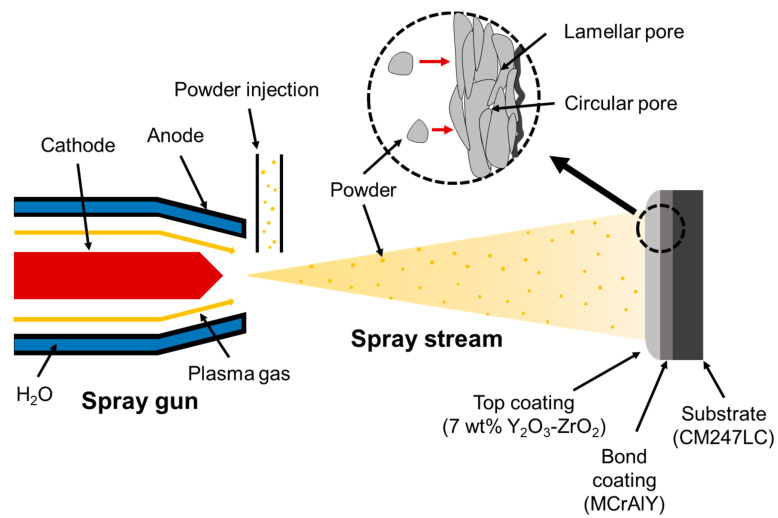
Schematic of APS system and TBC coating.

**Figure 2 materials-14-07801-f002:**
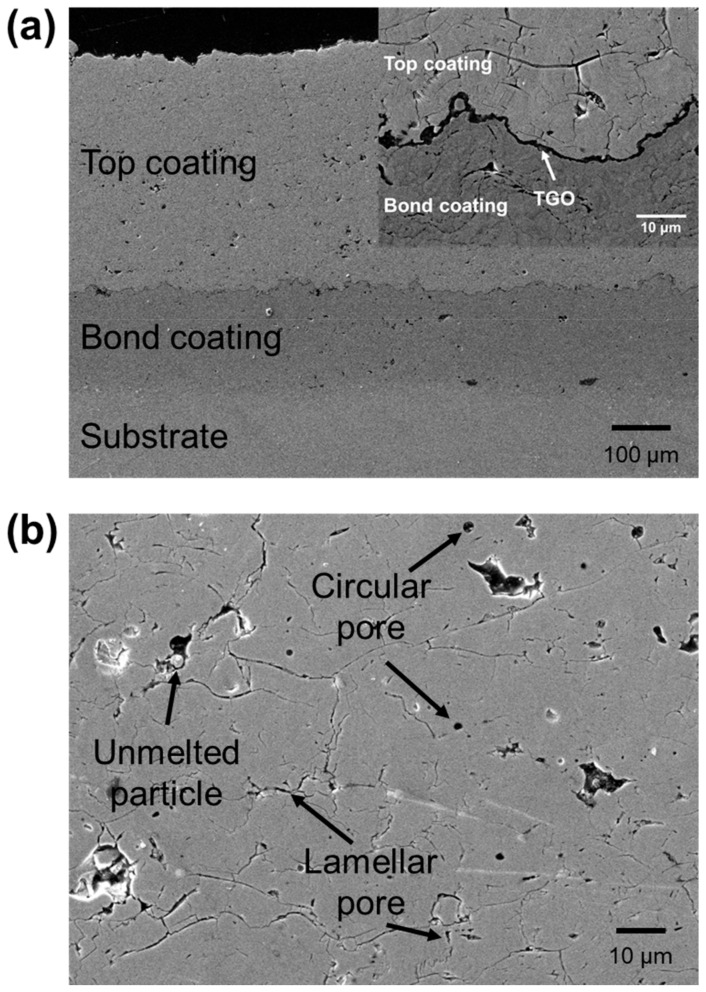
SEM image of TBC cross-section magnified at (**a**) × 200 and (**b**) × 1000. (inset) TGO between top coating and bond coating.

**Figure 3 materials-14-07801-f003:**
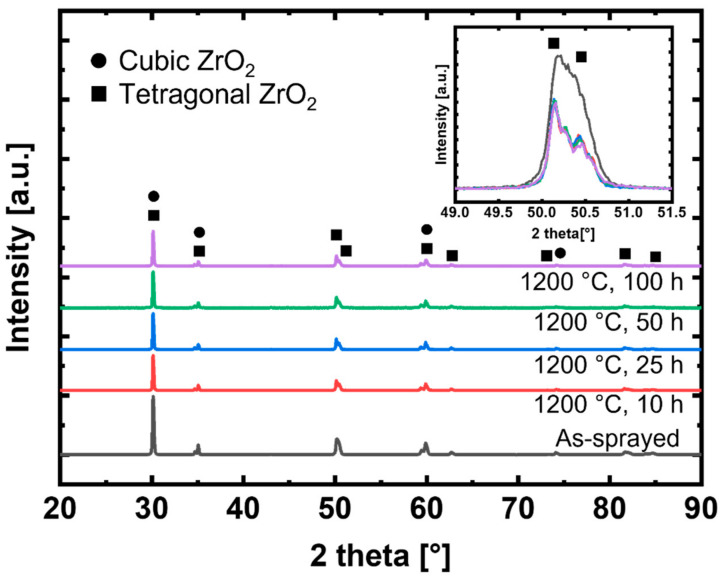
XRD spectra of as-sprayed and heat-treated 7YSZ coatings at 1200 °C. (inset) Magnified XRD spectra from 49 to 51.5°.

**Figure 4 materials-14-07801-f004:**
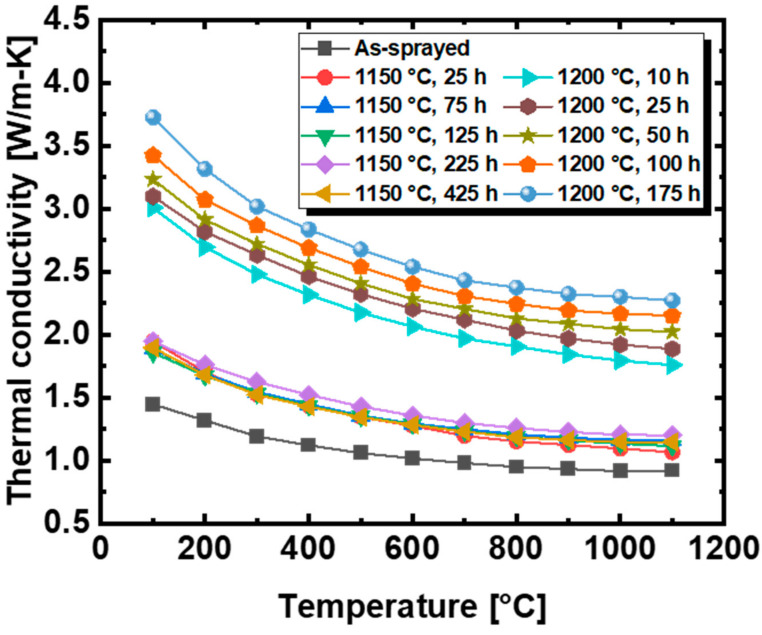
Thermal conductivity of as-sprayed and thermally aged YSZ.

**Figure 5 materials-14-07801-f005:**
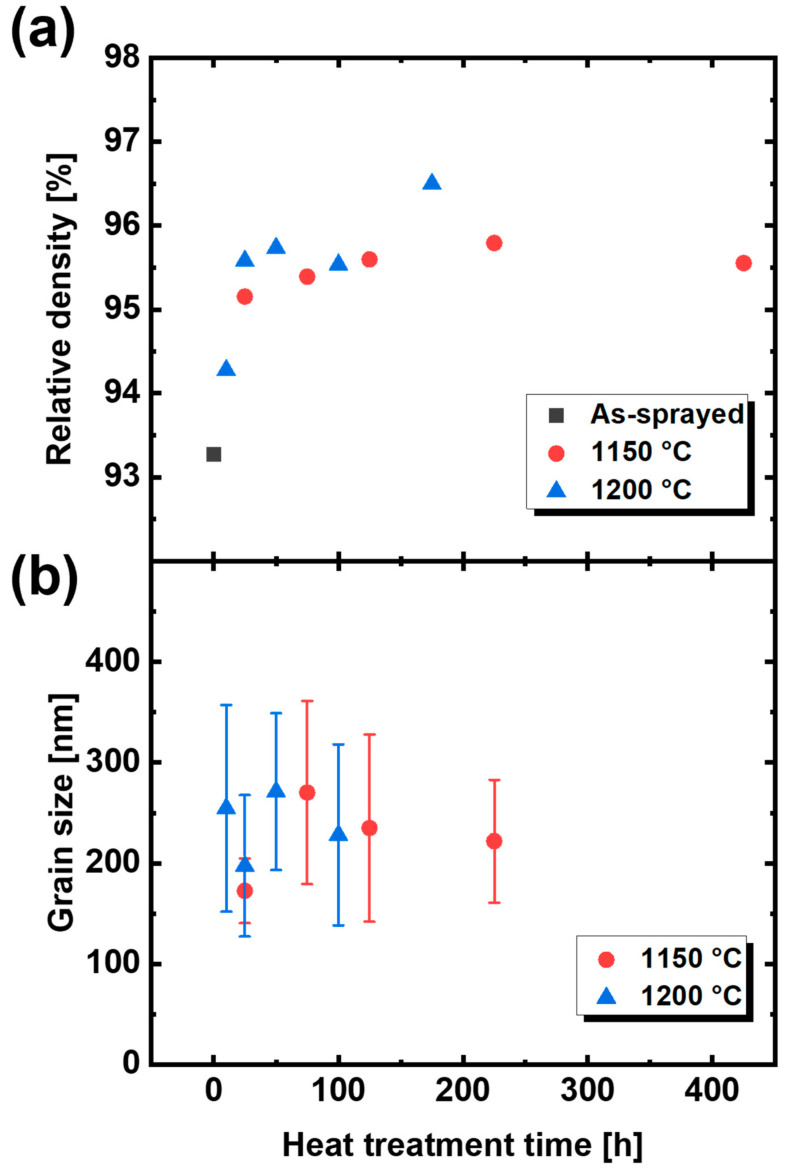
(**a**) Relative density and (**b**) grain size after thermal aging as a function of heat treatment time.

**Figure 6 materials-14-07801-f006:**
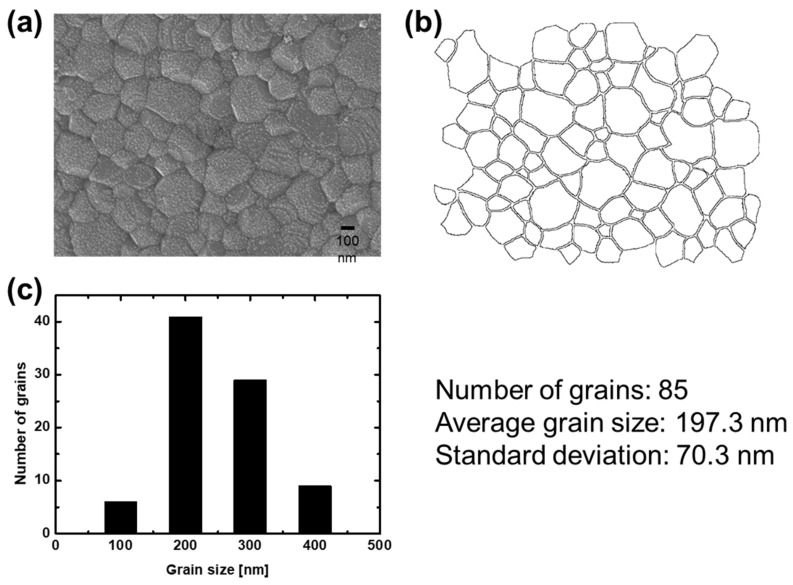
(**a**) SEM image of sample heat-treated at 1200 °C for 25 h, (**b**) image analysis and (**c**) histogram of grain size.

**Figure 7 materials-14-07801-f007:**
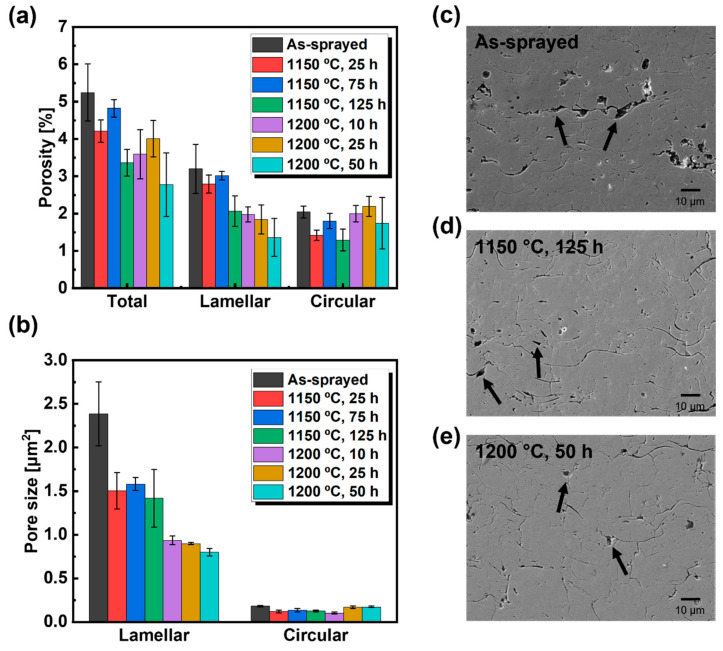
(**a**) Porosity and (**b**) pore size of YSZ samples. Cross-sectional SEM images of samples (**c**) as-sprayed and heat-treated at (**d**) 1150 °C for 125 h and (**e**) 1200 °C for 50 h. Arrows indicate lamellar pores.

## Data Availability

Not applicable.

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
