# Peer review of "Effect of Heat Treatment on Microstructure and Thermal Conductivity of Thermal Barrier Coating"

_materials, 2021, doi:10.3390/ma14247801_

Round 1

Reviewer 1 Report

The present article is well-written and well-articulated, where the authors have investigated the microstructural changes of TBC as a function of heat treatment. Some minor changes are required as follows:

  1. Details of the thermal sprayed powder was completely missing, which needs to be included.
  2. Page 105: It will be ‘investigated’ instead of ‘prepared’.

Author Response

We are sincerely grateful to the reviewers for their kind and very helpful comments. We have revised the manuscript according to the reviewers’ comments and suggestions as follows:

Point 1: Details of the thermal sprayed powder was completely missing, which needs to be included.

Response 1: According to reviewer’s comment, we have updated manuscript.

The material used for the top coating is a powder containing 7 wt% of Y2O3 based on ZrO2. The powder is agglomerated and sintered resulting in an average size of about 11 μm.

Point 2: Page 105: It will be ‘investigated’ instead of ‘prepared’.

Response 2: We have updated manuscript reflecting the reviewer’s comment.

Reviewer 2 Report

Thermal barrier coatings are widely applied in gas turbine parts due to the good energy efficiency. Authors studied the effect of heat treatment on the thermal conductivity. Some results have been carried out. However, the current form of this study cannot be acceptable. Some aspects as listed below:

  1. There are some errors of grammar in the manuscript. It is recommended to correct this document throughout.
  2. More details about APS should be given. What are the parameters of the APS system?
  3. In Fig,4, why the influence of heat treatment time on the thermal conductivity is not obvious at 1150 ゜C?
  4. More discussion about the grain sizes and thermal conductivity should be given.
  5. Why the coatings only heat-treated at 1150 and 1200 °C?

Author Response

We are sincerely grateful to the reviewers for their kind and very helpful comments. We have revised the manuscript according to the reviewers’ comments and suggestions as follows:

Point 1: There are some errors of grammar in the manuscript. It is recommended to correct this document throughout.

Response 1: The manuscript has been proofread by a native speaker.

Point 2: More details about APS should be given. What are the parameters of the APS system?

Response 2: According to reviewer’s comment, we have updated manuscript.

The top coating deposited with APS is sprayed on the substrate at 600 A and 80 V. The speed at which the top coating is sprayed on the base material is 300-600 mm/s. The spraying distance is about 76-127 mm.

Point 3: In Fig 4, why the influence of heat treatment time on the thermal conductivity is not obvious at 1150°C?

Response 3: We suspect that the change in thermal conductivity with increasing heat treatment time at 1150 °C is not obvious because the temperature is not high enough to affect the change in pore shape.

Point 4: More discussion about the grain sizes and thermal conductivity should be given.

Response 4: We have updated manuscript by reflecting this comments as below.

Phonons scattered by grain boundaries cause a decrease in thermal conductivity. Grain size is closely related to thermal conductivity. Overall, the grains had an average size of 200–300 nm and a standard deviation of 50–100 nm. The grain size and thermal conductivity of the coating after thermal aging increased compared to that of the as-sprayed coating. When the nanosized grains are on the same order as the mean free path of YSZ, the thermal conductivity is significantly reduced [21]. However, when the grain size of dense YSZ is between 70 and 400 nm, the change in thermal conductivity due to boundary scattering is insignificant [22].

[21] Kabacoff, L. In Thermally sprayed nano-structured thermal barrier coatings, NATO Workshop on Thermal Barrier Coatings, Aalborg, Denmark, AGARD, 1998; pp 143-149.

[22] Raghavan, S.; Mayo, M. J.; Wang, H.; Dinwiddie, R. B.; Porter, W. D., The effect of grain size, porosity and yttria content on the thermal conductivity of nanocrystalline zirconia. Scripta Materialia 1998, 39, (8), 1119-1125.

Point 5: Why the coatings only heat-treated at 1150 and 1200°C?

Response 5: We have chosen 1050-1200 °C as heat treatment since the range is close to the operating temperature. In particular, 1150 °C and 1200 °C were selected to understand the effect of high turbine inlet temperature for high efficiency.

We have updated manuscript.

Thermal aging, which simulates a coating being exposed to high temperatures for a long time, was performed in a furnace at 1150 and 1200 °C to examine the effect of changes in thermal conductivity at high temperatures. These thermal aging temperatures were chosen because they are close to the working temperature (1050–1200 °C) at which most YSZ is used [24].

[24] Huang, Y.; Hu, N.; Zeng, Y.; Song, X.; Lin, C.; Liu, Z.; Zhang, J., Effect of different types of pores on thermal conductivity of YSZ thermal barrier coatings. Coatings 2019, 9, (2), 138.

Reviewer 3 Report

Article can be considered to publishing in Materials after minor Revision.

The article Effect of Heat Treatment on Microstructure and Thermal Con-2 ductivity of Thermal Barrier Coating, shows the importance of the TBCs microstructure changes due to the sintering effect from heat treatment, which influences thermal conductivity of the coating (being the key feature in performance properties of TBC).

Please add in literature review – why thermal conductivity is specifically important in performance of TBC.

Is the bond coating formed as a results of reactions between components od top coating and substrate ? – lack of information – line 112-124.

Fig. 2a – “bond boating” change into “bond coating”

Line 268 – size of pores in micrometer^2?

Author Response

We are sincerely grateful to the reviewers for their kind and very helpful comments. We have revised the manuscript according to the reviewers’ comments and suggestions as follows:

Point 1: Please add in literature review – why thermal conductivity is specifically important in performance of TBC

Response 1: According to reviewer’s comment, we have updated manuscript.

The development of thermal barrier coatings (TBCs) has improved the durability of gas turbine components exposed to high temperatures and has improved the energy efficiency of gas turbines by increasing the turbine inlet temperature [1]. However, the temperature of the turbine inlet cannot be increased indefinitely due to the maximum operating temperature of the superalloy. When the surface temperature exceeds 900 °C, the nickel-based superalloy used as the base material is oxidized and corrodes [2]. Therefore, low thermal conductivity is essential for TBCs, to prevent oxidation and corrosion of superalloys at high turbine inlet temperatures. To lower the thermal conductivity of superalloys, several studies on structural changes [3, 4] and manufacturing using rare earths [5] have been conducted.

[1] Padture, N. P.; Gell, M.; Jordan, E. H., Thermal barrier coatings for gas-turbine engine applications. Science 2002, 296, (5566), 280-284.

[2] Kubacka, D.; Weiser, M.; Spiecker, E., Early stages of high-temperature oxidation of Ni-and Co-base model superalloys: A comparative study using rapid thermal annealing and advanced electron microscopy. Corrosion Science 2021, 191, 109744.

[3] Koolloos, M.; Marijnissen, G., Burner rig testing of" herringbone" EB-PVD Thermal Barrier Coatings. 2002.

[4] Lawson, K.; Nicholls, J.; Rickerby, D., Thermal conductivity and ceramic microstructure. ROLLS ROYCE PLC-REPORT-PNR 1998.

[5] Clarke, D. R.; Phillpot, S. R., Thermal barrier coating materials. Materials today 2005, 8, (6), 22-29.

Point 2: Is the bond coating formed as a results of reactions between components of top coating and substrate? – lack of information – line 112-124.

Response 2: Bond coating is not a results of reactions between top coating and substrate. It is a metal-based adhesive layer deposited between the substrate and the top coating. It is composed of Ni, Co, C, Al, Y and relieves the thermal stress caused by the difference in thermal expansion coefficient between top coating and the substrate. Also, it improves the oxidation resistance at high temperature.

According to reviewer’s comment, we have updated manuscript.

The bond coating plays a crucial role as it enhances the adhesion of the top coating [36], relieves the thermal stress caused by the difference in thermal expansion coefficient between the substrate and the ceramic top coating material [37], and improves oxidation resistance and hot corrosion resistance at high temperatures [38].

[36] Chen, G., Non-destructive evaluation (NDE) of the failure of thermal barrier coatings. In Thermal Barrier Coatings, Elsevier: 2011; pp 243-262.

[37] Avci, A.; Eker, A. A.; Eker, B., Microstructure and Oxidation Behavior of Atmospheric Plasma-Sprayed Thermal Barrier Coatings. In Exergetic, Energetic and Environmental Dimensions, Elsevier: 2018; pp 793-814.

[38] Zhou, C.; Song, Y., Oxidation and hot corrosion of thermal barrier coatings (TBCs). In Thermal Barrier Coatings, Elsevier: 2011; pp 193-214.

Point 3: Fig. 2a – “bond boating” change into “bond coating”

Response 3: We have updated manuscript accordingly.

Point 4: Line 268 – size of pores in micrometer^2?

Response 4: Since the size of the pores was analyzed from the cross-sectional image, μm, the unit of area, was used.

Reviewer 4 Report

The manuscript can be accepted after following corrections

  1. Introduction section: In the first paragraph, authors mentioned only 7YSZ, why not other compositions of YSZ?
  2. In the second paragraph of the introduction, authors discussed various manufacturing techniques for TBCs. However, the importance of this discussion is not mentioned as it does not suggest that one should use APS over other techniques.
  3. Page 2. Line 58: Authors stated that: “However, it is not known how heat treatment temperature or time affects the change in thermal conductivity?” and in line 83: “Numerous studies have investigated the relationship between heat treatment and thermal conductivity. It is known that thermal conductivity of TBCs increases after heat treatment.” These are two contradictory statements.
  4. Page 3, line 112-113: Please define what is ZRO-270-4, Amdry 9951 and CM247LC? Are these commercial names of the materials?
  5. Page 6, line 196-202: Authors discussed that the conductivity depends on the mean free path which depends on the interstitials, vacancies and grain boundaries. However, it has no relation with porosity, having porosity in any material will bring down the conductivity. Please correct the line 201-202. Also, it is not mentioned that how phono group velocity can be affected by microstructural features.
  6. In conclusion (line 306-307): Authors mentioned “Long-life coatings designed based on these results are expected to be applicable to high turbine inlet temperatures and in return further increase the efficiency of the turbine.” It implies that while designing the TBCs, one should retain certain porosity volume and pore size to avoid increasing the thermal conductivity, will it affect mechanical properties of the coating?

Author Response

We are sincerely grateful to the reviewers for their kind and very helpful comments. We have revised the manuscript according to the reviewers’ comments and suggestions as follows:

Point 1: Introduction section: in the first paragraph, authors mentioned only 7YSZ, why not other compositions of YSZ?

Response 1: According to reviewer’s comment, we have updated the introduction as the follows.

Introduction

An increase in the yttria content results in lower thermal conductivity of the TBC. However, the amount of yttria content should be within an appropriate range because it can affect the thermal cycle stability. The highest stability is shown when the yttria content is 7 wt% [6]. Therefore, 7YSZ is widely used as a material for TBC.

[6] Stecura, S., Optimization of the Ni-Cr-Al-Y/ZrO2-Y2O3 thermal barrier system. Adv. Cer. Mat.; (United States) 1986, Medium: X; Size: Pages: 68-76.

Point 2: In the second paragraph of the introduction, authors discussed various manufacturing techniques for TBCs. However, the importance of this discussion is not mentioned as it does not suggest that one should use APS over other techniques.

Response 2: We have updated manuscript accordingly.

EB-PVD, which is mainly used similar to APS, shows high stability, but has a complicated manufacturing process and exhibits relatively high thermal conductivity compared to APS [11]. On the other hand, APS manufacturing process is simple.

[11] Bernard, B.; Quet, A.; Bianchi, L.; Joulia, A.; Malié, A.; Schick, V.; Rémy, B., Thermal insulation properties of YSZ coatings: suspension plasma spraying (SPS) versus electron beam physical vapor deposition (EB-PVD) and atmospheric plasma spraying (APS). Surface and Coatings Technology 2017, 318, 122-128.

Point 3: Page 2. Line 58: Authors stated that: “However, it is not known how heat treatment temperature or time affects the change in thermal conductivity?” and in line 83: “Numerous studies have investigated the relationship between heat treatment and thermal conductivity. It is known that thermal conductivity of TBCs increases after heat treatment.” These are two contradictory statements.

Response 3: According to reviewer’s comment, we have updated manuscript.

Although the relationship between heat treatment and thermal conductivity has been studied, the exact cause of the change in thermal conductivity due to the heat treatment temperature and time is not known.

Point 4: Page 3, line 112-113: Please define what is ZRO-270-4, Amdry 9951 and CM247LC? Are these commercial names of the materials?

Response 4: ZRO-270-4, Amdry 9951 and CM247LC are the commercial names for thermal barrier coating powder, bond coating powder, and nickel-based superalloys.

Point 5: Page 6, line 196-202: Authors discussed that the conductivity depends on the mean free path which depends on the interstitials, vacancies, and grain boundaries. However, it has no relation with porosity, having porosity in any material will bring down the conductivity. Please correct the line 201-202. Also, it is not mentioned that how phonon group velocity can be affected by microstructural features.

Response 5: According to reviewer’s comment, we have updated manuscript.

Thermal conductivity is affected by heat capacity, phonon group velocity and mean free path [5]. The thermal conductivity due to lattice vibration can be expressed as k = 1/3 Cvvl, where Cv is the heat capacity, v is the phonon group velocity, and l is the mean free path of phonons. To change the thermal conductivity of a material, variations in Cv, v, and l are required. However, heat capacity is negligible above the Debye temperature. In the case of zirconia, the Debye temperature is 380 K; therefore, the changes in the specific heat above the Debye temperature are negligible [43]. Phonon group velocity can be expressed as v=(E/ρ)1/2 where E is the elastic modulus and ρ is the density of YSZ. The elastic modulus of YSZ increases as the heat treatment continues, and at high heat treatment temperatures the rate of increase is high [15, 44, 45]. Kim et al. [44] reported that the cause of the change in the elastic modulus of YSZ was due to pores or cracks inside the coating. There is no significant difference in the density of YSZ with respect to the heat treatment conditions as in Fig. 5(a). The mean free path is not affected at grain boundaries larger than several hundred nanometers [21]. On the other hand, the elastic modulus that affects the phonon group velocity is changed by pores or cracks. Therefore, we suspect that thermal conductivity after heat treatment is caused by changes in pores and cracks, which could affect the elastic modulus.

[5] Clarke, D. R.; Phillpot, S. R., Thermal barrier coating materials. Materials today 2005, 8, (6), 22-29.

[15] Thompson, J.; Clyne, T., The effect of heat treatment on the stiffness of zirconia top coats in plasma-sprayed TBCs. Acta materialia 2001, 49, (9), 1565-1575.

[21] Kabacoff, L. In Thermally sprayed nano-structured thermal barrier coatings, NATO Workshop on Thermal Barrier Coatings, Aalborg, Denmark, AGARD, 1998; pp 143-149.

[43] Slack, G. A., The thermal conductivity of nonmetallic crystals. Solid state physics 1979, 34, 1-71.

[44] Kim, D.; Park, K.; Kim, K.; Seok, C.-S.; Lee, J.; Kim, K., A method for predicting the delamination life of thermal barrier coatings under thermal gradient mechanical fatigue condition considering degradation characteristics. International Journal of Fatigue 2021, 151, 106402.

[45] Paul, S., Stiffness of plasma sprayed thermal barrier coatings. Coatings 2017, 7, (5), 68.

Point 5: In conclusion (line 306-307): Authors mentioned “Long-life coatings designed based on these results are expected to be applicable to high turbine inlet temperatures and in return further increase the efficiency of the turbine.” It implies that while designing the TBCs, on should retain certain porosity volume and pore size to avoid increasing the thermal conductivity, will it affect mechanical properties of the coating?

Response 5: Our study focused on how thermal conductivity changes as turbine operates at high temperature for long time. Reliability and mechanical strength should be researched separately.

Round 2

Reviewer 2 Report

Thermal barrier coatings are widely applied in gas turbine parts due to the good energy efficiency. Authors have revised the manuscript according to the commands. However, Fig.3 ,Fig.4 and Fig.5 should be improved.

Author Response

We have updated manuscript reflecting the reviewer's comment.
